# Encoding Robustness to Image Style via Adversarial Feature Perturbations

**Manli Shu**[1]     **Zuxuan Wu**[2]     **Micah Goldblum**[3]     **Tom Goldstein**[1]

[1] University of Maryland, College Park, USA
[2] Fudan University, Shanghai, China
[3] New York University, New York, USA

`manlis@cs.umd.edu, zxwu@fudan.edu.cn, goldblum@nyu.edu, tomg@cs.umd.edu`

## Abstract

Adversarial training is the industry standard for producing models that are robust to small adversarial perturbations. However, machine learning practitioners need models that are robust to other kinds of changes that occur naturally, such as changes in the style or illumination of input images. Such changes in input distribution have been effectively modeled as shifts in the mean and variance of deep image features. We adapt adversarial training by directly perturbing feature statistics, rather than image pixels, to produce models that are robust to various unseen distributional shifts. We explore the relationship between these perturbations and distributional shifts by visualizing adversarial features. Our proposed method, Adversarial Batch Normalization (AdvBN), is a single network layer that generates worst-case feature perturbations during training. By fine-tuning neural networks on adversarial feature distributions, we observe improved robustness of networks to various unseen distributional shifts, including style variations and image corruptions. In addition, we show that our proposed adversarial feature perturbation can be complementary to existing image space data augmentation methods, leading to improved performance. The source code and pre-trained models are released at `https://github.com/azshue/AdvBN`.

## 1 Introduction

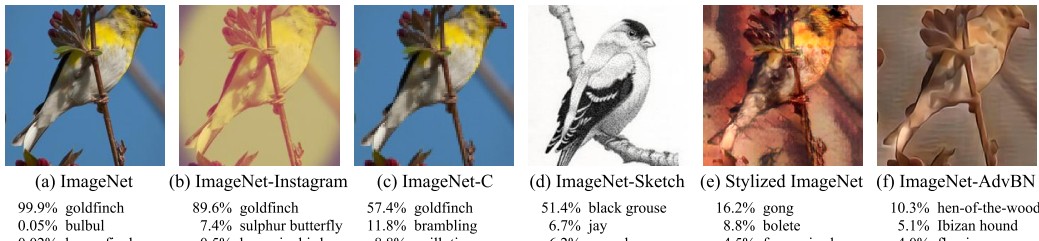

| (a) ImageNet | (b) ImageNet-Instagram | (c) ImageNet-C | (d) ImageNet-Sketch | (e) Stylized ImageNet | (f) ImageNet-AdvBN |
|---|---|---|---|---|---|
| 99.9% goldfinch | 89.6% goldfinch | 57.4% goldfinch | 51.4% black grouse | 16.2% gong | 10.3% hen-of-the-woods |
| 0.05% bulbul | 7.4% sulphur butterfly | 11.8% brambling | 6.7% jay | 8.8% bolete | 5.1% Ibizan hound |
| 0.02% house finch | 0.5% hummingbird | 8.8% guillotine | 6.2% coucal | 4.5% fox squirrel | 4.0% flamingo |

Figure 1: **Images from ImageNet variants along with classification scores by a pre-trained ResNet-50**. The image of column (a) is from ImageNet validation set. Dataset of column (d) is collected independently of the ImageNet dataset. Dataset of Column (f) is generated by our Adversarial Batch Normalization module. Details on how we generate column (f) can be found in Section 3.

Robust optimization for neural networks has been a major focus of recent research. A mainstream approach to reducing the brittleness of classifiers is *adversarial training*, which solves a min-max

35th Conference on Neural Information Processing Systems (NeurIPS 2021).

optimization problem in which an adversary makes perturbations to images to degrade network performance, while the network adapts its parameters to resist degradation [9, 18, 25]. The result is a hardened network that is no longer brittle to small perturbations to input pixels. While adversarial training makes networks robust to adversarial perturbations, it does not address other forms of brittleness that plague vision systems. For example, shifts in image style, lighting, color mapping, and domain shifts can still severely degrade the performance of neural networks [12].

We propose adapting adversarial training to make neural networks robust to changes in image style and appearance, rather than small perturbations at the pixel level. We formulate a min-max game in which an adversary chooses *adversarial feature statistics*, and network parameters are then updated to resist these changes in feature space that correspond to appearance differences of input images. This game is played until the network is robust to a variety of changes in image space including texture, color, brightness, *etc*.

The idea of adversarial feature statistics is inspired by the observation that the mean and variance of feature maps encode style information, and thus enable the transfer of style information from a source image to a target image through normalization [15, 40]. Unlike standard approaches that rely on feature statistics from auxiliary images to define an image style, we use adversarial optimization of feature statistics to prepare classifiers for the worst-case style that they might encounter.

We propose training with *Adversarial Batch Normalization* (AdvBN) layers. Before each gradient update, the AdvBN layer performs an adversarial feature shift by re-normalizing with the most damaging mean and variance. By using this layer in a robust optimization framework, we create networks that are resistant to various domain shifts representable by shifts in feature statistics. An advantage of this method is that it does not require additional auxiliary data from new domains. We show that robust training with AdvBN layer hardens classifiers against changes in image appearance and style [6, 45], as well as common image corruptions [12]. Besides classification, the effectiveness of AdvBN is also shown in the task of semantic segmentation, where it improves cross-domain generalization.

## 2 Related work

**Adversarial training.** Adversarial training and its variants [7, 25, 32, 49] have been widely studied for producing models that are robust to adversarial examples [26, 36] through solving min-max optimization problems. Besides defending against adversarial attacks, recent work has shown that adversarial training can be effectively applied to many other tasks [8, 17, 30, 53, 55]. Adversarial data augmentation [34, 42, 54] proposes to generate worst-case unseen domains using data augmentation and an adversarial loss, thus improving the generalization of neural networks. Another work [46] interprets the original formula of adversarial training as a type of data augmentation and reveals the distributional discrepancy between the feature representations of adversarial examples and clean data. The combination of adversarial training and feature statistics has been studied in previous work [24, 27], where it has been used to defend against adversarial attacks [27], or to generate feature distributions [24]. Our method differs from previous work in both the target of perturbation and the objective function. We craft adversarial feature distributions by directly perturbing the mean and variance of feature maps instead of through a variational auto-encoder [24], and our objective does not include regularization terms.

**Feature Perturbation.** Feature perturbation has been an effective approach to generate novel data distributions from a given source domain [14, 19, 22, 39]. Feature perturbations can be applied in different ways, such as adding spatial noise to the feature maps [37] or transforming feature maps using classical signal processing [22]. We will specifically discuss feature perturbation through re-normalization. While feature normalization [16, 40] is first proposed to accelerate neural network training [1], the mean and variance of deep feature representations have been shown to effectively capture image style information, and style transfer can be realized through feature re-normalization [15]. The idea of feature re-normalization has also been adopted to help neural networks adapt from the source domain to a target domain, using feature statistics obtained from the latter [21, 31]. Recent work [20] also proposes to use re-normalization for feature interpolation to improve the generalization capability of neural networks. Instead of re-normalizing features with statistics of other samples or from other domains, we simulate the worst-case scenario, encouraging models to be less sensitive to style information and thus to generalize better to images of varying appearances.

**Robustness to distribution shifts.** The definition of "distribution shifts" varies from one topic to another. For example, distribution shifts in the meta-learning literature[19, 37] often refer to the discrepancy in discriminatory feature distributions of novel tasks from different domains. Another definition is the subtle difference between training and testing data that are sampled from the same underlying distribution [38, 50]. This work mainly focuses on distribution shifts across image data that amount to major "style" discrepancy, including variations in illumination [6, 45], weather condition [29], and image quality degradation [12]. Various methods have been proposed to produce neural networks that generalize well to this type of distribution shift, including test time training [35], test time adaptation [43], training with noise [30], and novel network architectures [28], etc. Data augmentation is a popular method which is designed to increase the diversity of training data and prevent neural networks from over-fitting. Recent studies in data augmentation have proposed to use novel augmentation operations [6, 13, 48, 51], policy searching [3, 10, 23], and adversarial approaches [17, 42, 55], etc. However, the potency of data augmentation can be limited by the choice of applicable augmentations [2, 5]. Feature space augmentation through feature interpolation [20, 41], on the other hand, is not restricted to the family of image transformations. Our method also works in feature space, but instead of interpolating, we perturb feature statistics adversarially.

## 3 Adversarial Batch Normalization

We propose *Adversarial Batch Normalization* (AdvBN), a module that adversarially perturbs deep feature distributions such that the features confuse CNN classifiers. We iteratively compute adversarial directions in feature space by taking PGD steps on batch statistics. In the next section, we will train on these perturbed feature distributions in order to produce models robust to domain shifts.

Consider a pre-trained classification network, $g$, with $L$ layers. We divide $g$ into two parts, $g^{1,l}$ and $g^{l+1,L}$, where $g^{m,n}$ denotes layers $m$ through $n$. Now, consider a batch of data, $x$, with corresponding labels, $y$. Formally, the AdvBN module is defined by

$$\text{BN}_{\text{adv}}^{\delta}(x; g, l, y) = \delta'_{\sigma} \cdot \sigma(f) \cdot \left( \frac{f - \mu(f)}{\sigma(f)} \right) + \delta'_{\mu} \cdot \mu(f), \text{ where } f = g^{1,l}(x), \tag{1}$$

$$(\delta'_{\mu}, \delta'_{\sigma}) = \underset{(\delta_{\mu}, \delta_{\sigma})}{\arg \max} \mathcal{L}\left[ g^{l+1,L}\left( \delta_{\sigma} \cdot (f - \mu(f)) + \delta_{\mu} \cdot \mu(f) \right), y \right],$$
$$\text{subject to } \|\delta_{\mu} - 1\|_{\infty} \leq \epsilon, \|\delta_{\sigma} - 1\|_{\infty} \leq \epsilon, \tag{2}$$

where $\mathcal{L}$ is the cross-entropy loss, and the maximization problem is solved with projected gradient descent. Note that the feature input of $g^{l+1,L}$ is a simplified form of the AdvBN formulation in Eq (1), where the two $\sigma(f)$'s cancel out. Simply put, the AdvBN module is a PGD attack on batch norm statistics which can be inserted inside a network. $\delta_{\mu}, \delta_{\sigma}$ are vectors with length equal to the number of channels in the output of layer $l$, and we multiply by them entry-wise, one scalar entry per channel, similarly to Batch Normalization. Additionally, note that this module acts on a per-batch basis so that features corresponding to the same image are perturbed differently across training epochs as training samples are randomly shuffled between epochs during training.

In Eq (1), we re-normalize the feature by adding adversarial statistics $\delta_{\mu} \cdot \mu(f)$, rather than simply $\delta_{\mu}$, so that $\ell_{\infty}$ bounds and steps size do not need to depend on $\mu(f)$. Intuitively, we permit the mean of adversarial features to vary more when the clean features have a mean of higher magnitude.

**Visualizing feature shifts.** We adopt the VGG [33] based autoencoder from AdaIN [15], in which the encoder is the first few layers of a pre-trained VGG-19 image classification network. The decoder is trained to restore the input image from the output of the encoder. After we obtain an autoencoder that can perform the identity mapping on input images, we plug in an AdvBN module after the encoder. To compute cross-entropy loss for the AdvBN module, the remaining downstream layers of the aforementioned VGG-19 classifier are used, which takes the encoded feature as input and outputs the class prediction. The features perturbed by AdvBN are then fed into the decoder to create our visualizations.

In Figure 2, images with adversarial feature distributions exhibit differences in color, texture, and edges. We draw two major conclusions from these visualizations which highlight the adversarial properties of these domains. The first one concerns textures: CNNs have been shown to rely heavily

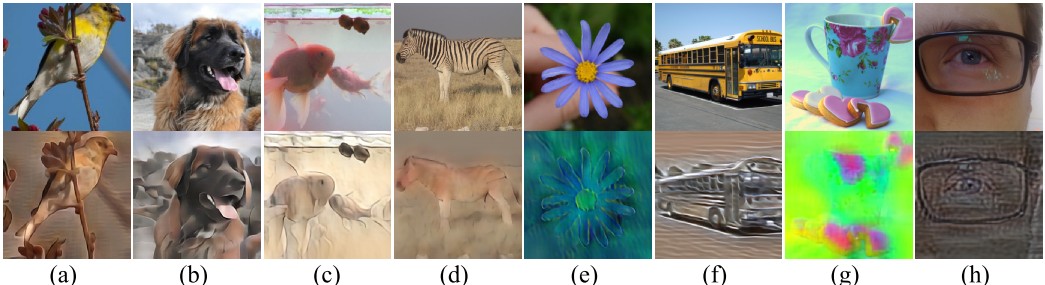

|       |       |       |       |       |       |       |       |
|-------|-------|-------|-------|-------|-------|-------|-------|
| (a)   | (b)   | (c)   | (d)   | (e)   | (f)   | (g)   | (h)   |

Figure 2: **Examples of ImageNet images with adversarial feature distributions shifted by Ad-vBN, visualized through a decoder.** For each pair, the original image is on the top.

on image textures for classification [6]. Images from the adversarial domain, on the other hand, have inconsistent textures across samples. For example, the furry texture of a dog is smoothed in Figure 2 (b), and the stripes disappear from a zebra in Figure 2 (d), whereas visible textures appear in (f) and (g) of Figure 2. The second conclusion pertains to color. Previous study [52] suggests that colors serve as important information for CNNs. In the adversarial domain, we find suppressed colors (Figure 2 (a), (c)) and unnatural hue (Figure 2 (e), (g)). Figure 3 illustrates how the appearance of reconstructed images shifts as adversarial perturbations to feature statistics become larger. See Appendix B for additional examples.

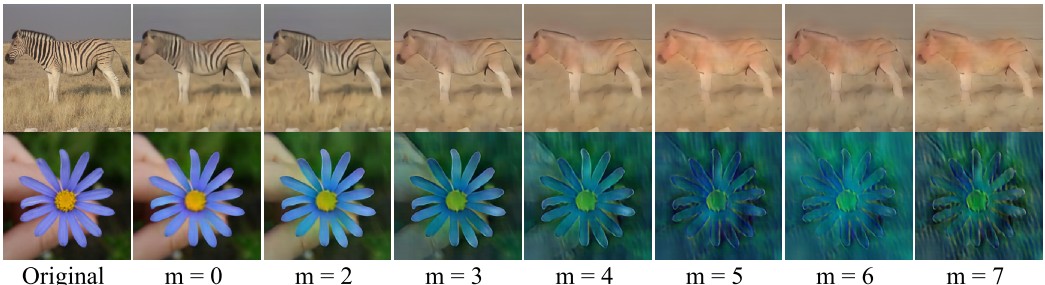

|          |       |       |       |       |       |       |       |
|----------|-------|-------|-------|-------|-------|-------|-------|
| Original | m = 0 | m = 2 | m = 3 | m = 4 | m = 5 | m = 6 | m = 7 |

Figure 3: **The effect of adversarial strength on visualized examples**. $m$ denotes the number of PGD steps. $m = 0$ corresponds to images reconstructed by our autoencoder without AdvBN.

We use this visualization technique to process the entire ImageNet validation set and denote it as ImageNet-AdvBN in Figure 1. By evaluating different methods on this dataset, we observe that performance on ImageNet-AdvBN is consistently degraded, which validates the adversarial property of features generated by AdvBN. Experiments concerning performance on ImageNet-AdvBN are listed in Appendix B.

## 4   Training with Adversarial Batch Normalization

In this section, we use the proposed AdvBN module to train networks on the perturbed features. The goal is to produce networks that generalize well to unseen domains while maintaining performance on the training distribution, all without having to obtain auxiliary data from new domains.

We start with a pre-trained model, $g = g^{l+1,L} \circ g^{1,l}$, and we fine-tune the subnetwork, $g^{l+1,L}$, on clean and adversarial features simultaneously. To this end, we solve the following min-max problem,

$$\min_{\theta} \mathbb{E}_{(x,y)\sim\mathcal{D}} \left[ \max_{\delta} \mathcal{L}(g_{\theta}^{l+1,L} \circ \mathrm{BN}_{\mathrm{adv}}^{\delta} \circ g^{1,l}(x), y) + \mathcal{L}(g_{\theta}^{l+1,L} \circ g^{1,l}(x), y) \right], \quad (3)$$

where $\mathcal{L}$ denotes cross-entropy loss, and $\mathcal{D}$ is the distribution of batches of size $n$. In order to maintain the network's performance on natural images, we adopt a similar approach to Xie et al. [46] by using auxiliary batch normalization in $g^{l+1,L}$ for adversarial features; we use the original BNs when propagating clean features, and we use auxiliary ones for adversarial features. See Algorithm 1 for a detailed description of our method.

Since we start with pre-trained models, we only need to fine-tune for 20 epochs, yielding improved robustness with little additional computation. Moreover, we only modify the parameters of later layers, so we do not need to backpropagate through the first half of the network. See Appendix C for an analysis of the training budget using our method. In the following section, we measure the performance, on several datasets, of our model fine-tuned using adversarial training with AdvBN.

---

**Algorithm 1:** Training with Adversarial Batch Normalization

---

**Input:** Training data, pretrained network $g = g_\theta^{l+1,L} \circ g^{1,l}$, learning rate $\alpha$, PGD bound $\epsilon$, and PGD step
      size $\tau$, loss function $\mathcal{L}$
**Result:** Updated network parameters, $\theta$, of subnetwork $g_\theta^{l+1,L}$
**for** *each training step* **do**
    Sample mini-batch $x$ with label $y$;
    Obtain feature map $f = g^{1,l}(x)$;
    Initialize perturbation: $\delta = (\delta_\mu, \delta_\sigma)$;
    Let $f_{adv} = f$;
    **for** *adversarial step = 1, ..., m* **do**
        $f_{adv} \leftarrow \delta_\sigma \cdot (f - \mu(f)) + \delta_\mu \cdot \mu(f)$;
        Update $\delta$:
        $\delta \leftarrow \delta + \tau \cdot sign(\nabla_\delta \mathcal{L}(g_\theta^{l+1,L}(f_{adv}), y))$;
        $\delta \leftarrow$ clip $(\delta, 1 - \epsilon, 1 + \epsilon)$;
    **end**
    $f_{adv} \leftarrow \delta_\sigma \cdot (f - \mu(f)) + \delta_\mu \cdot \mu(f)$;
    Update $\theta$ using gradient descent:
    $\theta \leftarrow \theta - \alpha \cdot \nabla_\theta \mathcal{L}(g_\theta^{l+1,L}(f_{adv}), y) + \mathcal{L}(g_\theta^{l+1,L}(f), y)$;
**end**
**return** $\theta$

---

## 5  Experiments

In section 5.1, we evaluate our method on image classification. We measure the generalization of models on ImageNet variant datasets that features different distributional shifts. We provide ablation studies of our method in Section 5.2. A feature divergence analysis is presented in Section 5.3 that validates the effectiveness of our method from a different perspective. In section 5.4, we evaluate our method on semantic segmentation. We conduct cross-domain evaluations on traffic scene datasets with different weather conditions and traffic scenes.

### 5.1  Evaluation on ImageNet Variants

In this section, we evaluate our method in the context of image classification on ImageNet variant datasets. We first evaluate the standalone performance of our method. We also include other baseline methods, including image space adversarial training adapted from the standard PGD adversarial training [25]. MoEx [20] is another related method that performs feature space augmentation through feature re-normalization. SIN [6] is trained on both Stylized ImageNet and original ImageNet, which uses AdaIN [15] as the style transfer method.

In addition, we examine AdvBN as a feature space augmentation method by showing its potential to be complementary to image space augmentation. We consider state-of-the-art data augmentation methods, including AutoAugment[3], Fast AutoAugment (AA)[23], CutMix[48], AugMix[13] and AdvProp[46]. We show that our method can further improve the generalization of models trained with advanced data augmentations. Results of all methods included in this section are based on the ResNet-50 model architecture. Our method also works well on other architectures, and we provide results in Appendix A.

**Implementation details.**    Our model begins with an ImageNet pre-trained ResNet-50 [11]. We insert the AdvBN module at the end of the $2^{nd}$ convolutional stage (`conv2_3`). We then fine-tune the model with our method following Algorithm 1 for 20 epochs. The learning rate starts at 0.001 and decreases by a factor of 10 after 10 epochs. Our batch size is set to 256. We use SGD with a momentum of 0.9 and weight decay coefficient $10^{-4}$. We search for the optimal number of adversarial

steps $m$ by increasing $m$ while fixing the step size to be $\tau = 0.2$, and we bound the perturbation with $\epsilon = m \cdot \tau - 0.1$. The optimal $m$ we find through this procedure is $m = 6$. When using AdvBN to improve a given data augmentation method, we apply this fine-tuning procedure on a model trained with the data augmentation, except for AA. Due to the absence of a pre-trained AA model, we manage to fine-tune a base model jointly with a fixed AA policy and AdvBN, and we compare it to a model solely fine-tuned with AA. All models trained with AdvBN that appeared in this subsection are obtained by following the same training routine and hyperparameter settings that we specified above.

**Datasets.** To measure the generalization ability of image classification models, we evaluate our models on four variants of ImageNet [4]:

- **ImageNet-C** [12] (under the Apache License 2.0) contains distorted images with 15 categories of common image corruption applied, each with 5 levels of severity. Performance on this dataset is measured by mean Corruption Error (mCE), the average classification error over a total of 75 combinations of corruption type and severity level, weighted by their difficulty.
- **ImageNet-Instagram (ImageNet-Ins.)** [45] is composed of ImageNet images processed with a total of 20 different Instagram aesthetic image filters. Filters are applied separately, and the dataset contains 20 sub-datasets, each corresponding to one type of image filter.
- **ImageNet-Sketch** [44] (under the MIT License) is a dataset of black and white sketches. The dataset includes 50,000 images in total, falling into 1,000 ImageNet categories, with 50 images per category. Images in this dataset are collected independently from the original ImageNet validation set through Google Image queries. Details concerning the construction of this dataset can be found in Section 4.4 of Wang et al. [44].
- **Stylized ImageNet (ImageNet-Style)** [6] (under the MIT License) consists of images from the ImageNet dataset, each stylized using AdaIN [15] with a randomly selected painting. Textures and colors of images in this dataset differ heavily from the originals.

Table 1: **Evaluation on ImageNet variants**. All methods are implemented based on ResNet-50. Performance on ImageNet-C is measured by mean Corrupted Error (mCE)[13].

| Method | ImageNet-C mCE $\downarrow$ | ImageNet-Ins. Top-1 acc. $\uparrow$ | ImageNet-Sketch.. Top-1 acc.$\uparrow$ | ImageNet-Style Top-1 acc.$\uparrow$ |
|---|---|---|---|---|
| Standard Training | 76.7 | 67.2 | 24.1 | 7.4 |
| Adv. Training | 73.7 | 68.2 | 25.3 | 9.1 |
| MoEx (w/ Cutmix) | 74.8 | 70.0 | 24.0 | 5.0 |
| SIN | 73.8 | 68.5 | 26.9 | 10.4 |
| AdvBN | 72.7 | 69.5 | 27.9 | 11.9 |
| AdvProp | 70.7 | 69.2 | 18.0 | 9.0 |
| AdvProp + AdvBN | 69.5 | 69.3 | 28.7 | 12.6 |
| Cutmix | 74.7 | 70.3 | 23.8 | 5.3 |
| Cutmix + AdvBN | 72.1 | 70.9 | 27.2 | 8.2 |
| AutoAugment*(AA) | 72.1 | 70.1 | 26.7 | 8.2 |
| AA + AdvBN | 68.6 | 71.1 | **30.3** | **14.1** |
| Fast AA | 68.7 | 71.1 | 27.2 | 8.3 |
| Fast AA + AdvBN | 68.7 | **71.3** | 28.6 | 11.4 |
| Augmix | 68.4 | 70.4 | 28.5 | 11.2 |
| Augmix + AdvBN | **64.6** | 71.1 | 28.7 | 13.6 |

**Model details.** Models trained using other methods that we include in this section are ResNet-50 models released by the authors of the original work, except for methods of which an official ResNet-50 is not available. The released MoEx model is trained collaboratively with CutMix. The "Adv. Training" baseline is adapted from PGD adversarial training, for which we adopt auxiliary batch normalization to alleviate the performance degradation on non-adversarial images, and the model is obtained through fine-tuning on a standard trained model. AdvProp does not provide a ResNet-50 model, and we use the open source implementation[1], and our reproduced model matches the accuracy reported in the aforementioned implementation. Our reported performance using AA, denoted as

---

[1] https://github.com/tingxueronghua/pytorch-classification-advprop (MIT License)

"AA*", is obtained through fine-tuning a pre-trained base model for the same number of epochs as AdvBN, using a fixed set of augmentation operations found by AA, which is included as a reference to the performance of "AdvBN + AA". The augmentation policy that we use is from the original work and can be found in the open source implementation[2].

**Results.**    In Table 1, we evaluate the performance of models on the four variant datasets. The standalone AdvBN improves the generalization of a standard model on every dataset and is competitive with alternative methods. Additionally, we find that our method is complementary to input space data augmentation, consistently boosting the performance of state-of-the-art data augmentation methods. Note that our model has auxiliary BN layers, so its performance on the original ImageNet is well maintained, as will be shown in the next subsection. Details concerning inference are in Appendix A.

## 5.2    Ablation Study

**Where should the AdvBN module be placed within a network?**    The proposed AdvBN module can be inserted after any layer in a network. In this section, we try AdvBN after other layers, namely `conv3_4` and `conv4_6`. For the ablation study, all of our models are obtained by following the same fine-tuning setting found in subsection 5.1, but with a fixed AutoAugment policy as data augmentation. In Table 2, we observe that `conv4_6` yields the worst performance among all three ImageNet variants, indicating that using AdvBN at such deep layer is not as helpful as at shallower layers. We hypothesize two possible explanations for this phenomenon: (1) there are fewer trainable parameters when only very deep layers are fine-tuned; (2) features are more abstract at deeper layers, and perturbing these high-level features can lead to extremely chaotic feature representations that are harmful for classification.

Table 2: **Ablation studies on the positioning of AdvBN**. The base model is ResNet-50.

| Model | ImageNet top1 acc. ↑ | ImageNet-C mCE. ↓ | ImageNet-Ins. top1 acc. ↑ | ImageNet-Sketch. top1 acc. ↑ | ImageNet-Style top1 acc. ↑ |
|---|---|---|---|---|---|
| Base model | 76.1 | 76.7 | 67.2 | 24.1 | 7.4 |
| $l = $ `conv2_3` | 76.5 | 68.6 | 71.1 | 30.3 | 14.1 |
| $l = $ `conv3_4` | 76.2 | 70.0 | 70.2 | 33.2 | 19.5 |
| $l = $ `conv4_6` | 75.3 | 75.0 | 68.5 | 26.1 | 11.0 |

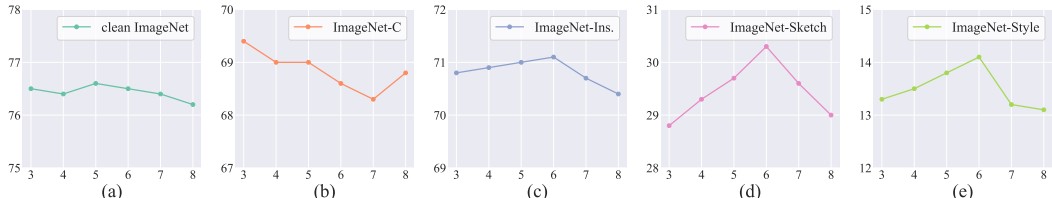

Figure 4: **The effects of adversarial strength.** The y-axis of (b) ImageNet-C is the mean Corrupted Error (mCE), and the others are top-1 accuracies. X-axes are the number of PGD steps $m$.

**Adversarial strength.**    The strength of the adversarial attack in the adversarial training framework has a major impact on model performance [25]. We test a range of PGD parameters to demonstrate how the strength of AdvBN affects model performance. We measure strength by the perturbation number of PGD steps $m$, where we fix $\tau$ to be 0.2 for all settings and fix the perturbation bound $\epsilon = [m \cdot \tau] - 0.1$ for different $m$'s.

Results concerning the impact of adversarial strength are shown in Figure 4. We can see that the clean accuracy on ImageNet decreases as the number of steps $m$ grows. On the other datasets, we can observe a turning point, where the performance reaches optimality. This behavior is expected because small perturbations cause small changes to features which may help maintain the clean accuracy but cannot help improve a model's generalization to other domains; overly large perturbations are also less beneficial as the resulting features can be too noisy.

---

[2]`https://github.com/DeepVoltaire/AutoAugment` (MIT License)

## 5.3 Feature Divergence Analysis

We compare the features extracted by our network to those of a standard ResNet-50 trained on ImageNet. Following Pan et al. [28], we model features from each channel using a normal distribution with the same mean and standard deviation, and we compute the symmetric KL divergence between the corresponding distributions on the two datasets ($A$ and $B$). For two sets of deep features, $F_A$ and $F_B$, each with $C$ channels, the divergence $D(F_A||F_B)$ is computed using the formula,

$$D(F_A||F_B) = \frac{1}{C}\sum_{i=1}^{C}(KL(F_A^i||F_B^i) + KL(F_B^i||F_A^i)), \tag{4}$$

$$KL(F_A^i||F_B^i) = \log\frac{\sigma_B^i}{\sigma_A^i} + \frac{\sigma_A^{i^2} + (\mu_A^i - \mu_B^i)^2}{2\sigma_B^{i^2}} - \frac{1}{2}, \tag{5}$$

where $F^i$ denotes the features of $i$-th channel with mean $\mu^i$ and standard deviation $\sigma^i$.

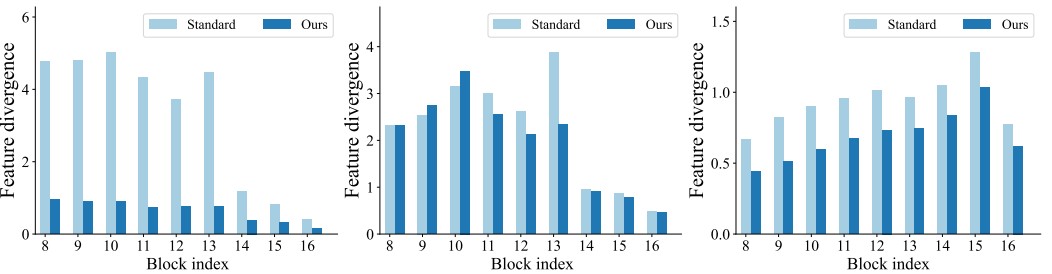

(a) ImageNet vs. ImageNet-Ins.    (b) ImageNet vs. ImageNet-Sketch    (c) ImageNet vs. ImageNet-Style

Figure 5: **Feature divergence between pairs of datasets.** Features are extracted by a standard and an AdvBN fine-tuned ResNet50.

In Figure 5, we compare the baseline model with our own on three pairs of datasets in the fine-tuned layers. Since ImageNet-Instagram contains 20 filter versions, we use the "Toaster" filter found in [45] to cause the sharpest drop in classification performance.

Feature divergence in our network trained with AdvBN is substantially smaller in the deeper layers of the fine-tuned subnetwork. In other words, the distribution of deep features corresponding to shifted domains is very similar to the distribution of deep features corresponding to standard ImageNet data. The small divergence between feature representations explains the effectiveness of AdvBN from a different angle and explains why our model generalizes well across datasets.

## 5.4 Generalization on Semantic Segmentation

**Datasets.**    We present domain generalization results on the Synthia video sequences dataset[3] [29], consisting of multiple sub-datasets featuring traffic situations under different weather, illumination, and season conditions. We conduct experiment on 10 sub-datasets that include two different road scenes: "Highway" and "New York-like City", each one with 5 different domain shifted variants: "dawn", "fog", "night", "spring" and "winter". Figure 6 shows sample images from each of the 10 sub-datasets. We use the left-front view images of each sub-dataset, and split dataset by randomly selecting 900 images for training and 500 for validation. We separately train two sets of models on the "Highway/ Dawn" and "New York-like City/ Spring" datasets and evaluate them on all the 10 sub-datasets within the two road scenes.

**Implementation details.**    The segmentation model we use is a ResNet-50 based segmentation network with dilated convolutions [47]. Our baseline models are trained for 80 epochs following the training protocol from [28]. We apply AdvBN by placing it after layer `conv2_3` of the baseline model and fine-tuning on a given sub-dataset for 30 epochs, with adversarial training parameters $\tau = 0.2$, $\epsilon = 0.5$, and 3 repeats.

---

[3]`http://synthia-dataset.net/`, subject to Attribution-NonCommercial-ShareAlike 3.0.

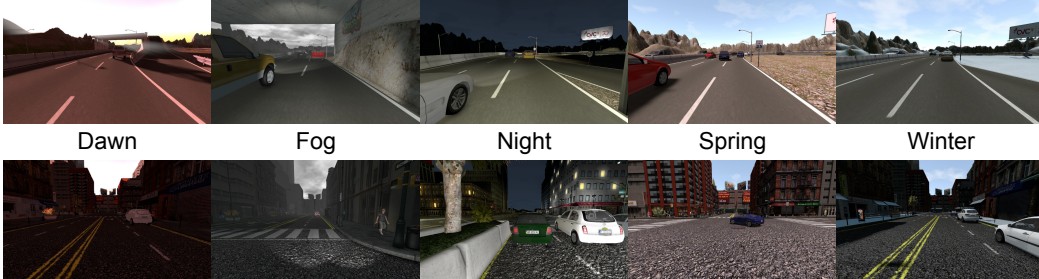

Figure 6: **Example images from the Synthia video sequences dataset**. The top row contains traffic scenes of the "Highway" subset, and the bottom row is the "New York-like City".

Table 3: **Segmentation results on the Synthia dataset**. Evaluation metric is mean IoU (Intersection over Union). The first column denotes the dataset used for training.

| | | New York-like City | | | | | Highway | | | | |
| --- | --- | --- | --- | --- | --- | --- | --- | --- | --- | --- | --- |
| | | Dawn | Fog | Night | Spring | Winter | Dawn | Fog | Night | Spring | Winter |
| NY.Like C./ Spring | baseline | 52.7 | 49.5 | 49.7 | 65.9 | 48.2 | 18.6 | 21.0 | 16.9 | 21.6 | 15.3 |
| | + Adv. Training | 54.6 | 50.7 | 50.2 | 65.8 | 49.7 | 21.2 | **28.0** | 20.4 | 26.3 | 21.4 |
| | AdvProp | 52.3 | 48.9 | 48.0 | **71.9** | 48.2 | 18.9 | 22.0 | 13.8 | 24.3 | 17.5 |
| | MoEx | 54.7 | 53.2 | 51.8 | 71.1 | 49.0 | 21.3 | 24.3 | 19.4 | 27.6 | 18.7 |
| | + AdvBN | **57.5** | **55.1** | **55.4** | 66.5 | **52.7** | **23.8** | 26.6 | **25.9** | **29.8** | **23.5** |
| Highway/ Dawn | baseline | 32.6 | 29.0 | 25.4 | 24.2 | 24.8 | 64.2 | 55.5 | 53.1 | 59.0 | 49.2 |
| | + Adv. Training | 33.5 | 30.7 | 27.9 | 27.5 | 26.7 | 64.0 | 56.4 | 54.0 | 59.5 | 50.3 |
| | AdvProp | 30.8 | 24.1 | 20.3 | 23.2 | 21.4 | **64.6** | 53.5 | 47.2 | 59.0 | 47.6 |
| | MoEx | 32.0 | 27.5 | 27.6 | 29.6 | 26.7 | **64.6** | 57.2 | **57.0** | 61.0 | 51.1 |
| | + AdvBN | **34.0** | **31.6** | **29.6** | **30.7** | **29.1** | 64.5 | **57.2** | 56.4 | **61.2** | **53.2** |

We also include the results of several alternative methods from Section 5.1. For the image space adversarial training, we adopt the same training settings as we do for AdvBN, except for the adversarial training parameters (perturbation size and bound). We find the optimal perturbation size for adversarial training through grid search and report the best results. The optimal hyperparameters we find are $\tau = 1$, $\epsilon = 1$. Auxiliary BNs are not used for either method at inference time. For MoEx and AdvProp, we train them for 110 epochs to match the number of optimization steps with our AdvBN fine-tuned model. Note that we do not include other data augmentation methods from Section 5.1 because they are originally designed for the image classification problems, which include operations that can be tricky to be applied to dense prediction problems like semantic segmentation.

We evaluate the performance of semantic segmentation using the mean IoU (Intersection over Union) metric. In table 3, over 20 source-target domain pairs, we show that AdvBN achieves the best cross-domain generalization performance. Our method also improves in-domain performance over standard training, while AdvProp achieves the highest performance under in-domain settings. Results in this section are consistent with our observation on image classification in Section 5.1.

## 6 Conclusion and Discussion

Our work studies how perturbing feature statistics simulate distribution shifts in image data. We find that fine-tuning on images with adversarially shifted feature distributions improves a model's robustness towards various domain shifts without using auxiliary data. As AdvBN operates purely in feature space, it is complementary to existing input space data augmentation methods, and can further improve the generalization of state-of-the-art methods. Future work will be to adapt our method to tasks beyond vision. It is known that adversarial perturbations in input space boost performance for language [55] and graph [17] models, and these data modalities may benefit from more structured feature-space perturbations.

**Limitations and Impact.** While AdvBN can offer impactful improvements for domain generalization, it may in some cases trade off performance on non-shifted in-distribution testing data. Moreover, real-world datasets and distributional shifts vary dramatically, and practitioners should be cautious rather than expecting that the performance seen on benchmark datasets, such as ImageNet variants, will translate to performance boosts in their own settings.

## Acknowledgements

Shu and Goldstein were supported by the ONR MURI program, with additional support provided by DARPA GARD. Wu was supported by NSFC under Grant No. 62102092.

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
