# A  Additional results and experiment details

## A.1  Detailed results on ImageNet-C

In this section, we provide a detailed version of the results shown in our experiment section concerning the ImageNet-C dataset, which technically contains a total of 75 variants of the ImageNet dataset. The 75 variants fall into 15 categories of corruption; each category presents 5 gradually increasing degrees of severity, where "degree=1" denotes the lowest degree of severity.

Table 4: **mean Corrupted Error (mCE) of each corruption category in ImageNet-C**.

| Network | Clean | Noise | | | Blur | | | | Weather | | | | Digital | | | | mCE |
|---|---|---|---|---|---|---|---|---|---|---|---|---|---|---|---|---|---|
| | | Gauss. | Shot | Impulse | Defocus | Glass | Motion | Zoom | Snow | Frost | Fog | Bright | Contrast | Elastic | Pixel | JPEG | |
| Base model | 23.9 | 80 | 82 | 83 | 75 | 89 | 78 | **80** | 78 | 75 | 66 | 57 | 71 | 85 | 77 | 77 | 76.7 |
| AdvBN | **23.0** | **75** | **76** | **77** | **70** | **85** | **75** | **80** | **74** | **71** | **63** | **54** | **66** | **82** | **71** | **72** | **72.7** |

Table 5: **Raw error of each subset in ImageNet-C** .

| Model | Corruption | Degree 1 2 3 4 5 | Corruption | Degree 1 2 3 4 5 | Corruption | Degree 1 2 3 4 5 | Corruption | Degree 1 2 3 4 5 |
|---|---|---|---|---|---|---|---|---|
| AugMix + AdvBN | Blur-Defocus | 35 40 50 62 74
34 39 51 64 74 | Blur-Glass | 39 50 74 78 84
38 49 74 79 86 | Blur-Motion | 28 33 42 58 71
29 36 50 69 80 | Blur-Zoom | 38 45 49 57 65
42 51 57 65 73 |
| AugMix + AdvBN | Weather-Snow | 39 59 57 69 77
40 59 56 68 75 | Weather-Frost | 35 50 62 64 71
34 49 60 62 69 | Weather-Fog | 37 42 52 58 75
34 40 48 55 72 | Weather-Bright | 25 26 29 33 40
24 25 28 32 37 |
| AugMix + AdvBN | Digital-Contrast | 29 33 39 59 85
29 33 41 63 87 | Digital-Elastic | 31 53 37 48 71
31 53 38 50 75 | Digital-Pixel | 30 32 41 53 60
30 31 40 53 58 | Digital-JPEG | 32 35 37 43 52
31 34 36 41 49 |
| AugMix + AdvBN | Noise-Gauss. | 32 40 55 76 94
31 37 48 64 84 | Noise-Shot | 33 42 55 77 88
32 39 49 69 81 | Noise-Impulse | 36 46 56 79 95
37 44 51 67 84 | | |

In Table 4, we list the mCE of each corruption category. The mCE of a given category is calculated by taking the average of the 5 corruption errors corresponding to the 5 corruption degrees of the given category, and then normalize the mean value with a constant. The constant differs between corruption categories, which reflects the difficulty of a given corruption type. For detailed formulations, please refer to the official ImageNet-C repository[4]. From the results, we can see that AdvBN alone can improve the baseline model on all corruption types. In Table 5, we list the raw error rate on each sub-dataset in ImageNet-C. The two models in this table are the AugMix model and AugMix model fine-tuned with AdvBN respectively.

## A.2  Other architectures

We apply our method to other network architectures and evaluate on the task of image classification. Datasets in Table 6 are the same as in Table 1. We apply AdvBN using the same setting as introduced in Section 5.1, by fine-tuning a pre-trained model for 20 epochs using SGD optimizer. For DenseNet-121, we place the AdvBN layer after the first block, and we use 6 PGD steps with stepsize $\tau = 0.2$, $\epsilon = 1.1$. For the EfficientNet, we place the AdvBN layer after the second block, and we use 3 PGD steps with stepsize $\tau = 0.2$, $\epsilon = 0.5$.

Table 6: **Applying AdvBN to other architectures**.

| Architecture | ImageNet-C mCE. ↓ | ImageNet-Ins. Top-1 acc. ↑ | ImageNet-Sketch Top-1 acc. ↑ | ImageNet-Style Top-1 acc. ↑ |
|---|---|---|---|---|
| DenseNet-121 | 73.4 | 66.6 | 24.3 | 7.9 |
| + AdvBN (w/ AA) | **70.4** | **69.3** | **28.6** | **15.5** |
| EfficientNet-B0 | 72.1 | 69.7 | 26.7 | 12.5 |
| + AdvBN (w/ AA) | **68.7** | **71.3** | **27.4** | **15.7** |

---

[4] https://github.com/hendrycks/robustness

### A.3 AdvBN at inference time

Models containing Batchnorm layers, such as ResNet, will have two sets of BN statistics in deeper layers after being fine-tuned by our method, because we use auxiliary BN [46] for propagating adversarially perturbed features. The auxiliary BN in our model stores the mean and standard deviation of the adversarial feature, which is very different from the feature statistics of the original data.

Intuitively, when testing on data whose distribution is "close" to the training data, using the main BN (as compared to the auxiliary BN) in a model would be more favorable than using the auxiliary BN, and vice versa. In this work, we take a naive measurement for the "closeness" of an ImageNet variant to the original ImageNet by comparing a model's accuracy on the two datasets. For example, on ImageNet-Instagram, the accuracy of a standard baseline is not degraded much from that on the original ImageNet validation set, and we consider it to be "close" to the original data. Following this rule, at the inference time, we use the main BN on the original ImageNet, ImageNet-Instagram and ImageNet-C, and use auxiliary BN on ImageNet-Sketch and ImageNet-Style. A future direction is to improve the measurement of the distance between test samples and the training data using unsupervised techniques. To demonstrate the discrepancy of the main and auxiliary BN statistics in our model, we include full results of using both statistics in Table 7.

Table 7: **Evaluation using main and auxiliary batch normalization statistics respectively**.

| Method | ImageNet-C mCE ↓ | ImageNet-Ins. Top-1 acc. ↑ | ImageNet-Sketch.. Top-1 acc.↑ | ImageNet-Style Top-1 acc.↑ |
|---|---|---|---|---|
| Standard Training | 76.7 | 67.2 | 24.1 | 7.4 |
| AdvBN (w/ main BN) | 72.7 | 69.5 | 26.4 | 9.0 |
| AdvBN (w/ aux. BN) | 72.4 | 68.5 | 27.9 | 11.9 |

## B  ImageNet-AdvBN Dataset

### B.1  Creation of the ImageNet-AdvBN dataset

We process the entire ImageNet validation set using the visualization technique introduced in Section 3. We consider two encoder architectures: one is the VGG-19 encoder we use for visualization, another consists of layers of a ResNet-50 up to `conv2_3`. Both encoders are paired with the same decoder architecture from Huang and Belongie [15]. The resulting datasets, denoted by ImageNet-AdvBN-VGG and ImageNet-AdvBN-ResNet respectively, contain 50000 images each. The data we synthesize for testing other models is generated using these autoencoders that contain the AdvBN module but on ImageNet validation data. AdvBN is conducted with 6 steps, stepsize $= 0.20$, $\epsilon = 1.1$, and a batch size of 32. We do not shuffle the ImageNet validation data when generating these batches.

### B.2  Classification on ImageNet-AdvBN

Table 8 shows the classification performance of various models on the two ImageNet-AdvBN variants, denoted as IN-Adv-VGG and IN-Adv-ResNet respectively. Models in Table 8 are the same ResNet-50 models we use in section 5.1, where we give the details of each model. The significantly degraded accuracy on our generated dataset indicates the adversarial property of our method. We also test these models on ImageNet images reconstructed using our autoencoders, denoted as VGG Reconstructed and ResNet Reconstructed, for each autoencoder. The performance gap between ImageNet-AdvBN and Reconstructed ImageNet indicates that the degradation on ImageNet-AdvBN is not solely caused by the reconstruction loss due to the autoencoders we use.

### B.3  Additional Example Images

We include more images from ImageNet-AdvBN-VGG in this section. Example images in Figure 7 are randomly chosen. We do not include the ImageNet-AdvBN-ResNet, because the resulting images are mostly in extreme contrast with small textures that are hard to observe. It is possible that features output from ResNet based encoders are more sensitive to AdvBN perturbations; another explanation is that the features we extract from ResNet-50 are relatively shallow features compared to their VGG counterparts.

Table 8: **Classification accuracy on ImageNet-AdvBN and reconstructed images**. Models of all methods are implemented based on ResNet-50 and trained on the original ImageNet training set. IN-Adv-VGG and IN-Adv-ResNet are two ImageNet-AdvBN datasets generated using different auto-encoders. VGG-reconstructed and ResNet-reconstructed are two datasets generated using the same auto-encoder as their AdvBN counterparts but without feature perturbation.

| Method | ImageNet Top-1 acc. ↑ | IN-Adv-VGG Top-1/ Top-5 acc. ↑ | VGG Reconstructed Top-1/ Top-5 acc. ↑ | IN-Adv-ResNet Top-1/ Top-5 acc. ↑ | ResNet Reconstructed Top-1/ Top-5 acc. ↑ |
|---|---|---|---|---|---|
| Standard Training | 76.1 | 1.6/ 4.7 | 45.8/ 70.6 | 0.4/ 1.3 | 65.7/ 86.9 |
| MoEx (w/ Cutmix) | 79.1 | 1.0/ 2.9 | 40.2/ 63.8 | 0.3/ 1.1 | 65.7/ 86.8 |
| Adv. Training | 76.6 | 2.0/ 5.5 | 48.1/ 72.5 | 0.5/ 1.5 | 68.0/ 88.3 |
| AdvBN | 77.0 | 4.7/ 11.9 | 46.8/ 71.4 | 1.7/ 4.0 | 67.2/ 87.9 |
| AdvProp | 77.4 | 1.6/ 4.5 | 53.1/ 76.6 | 0.3/ 0.9 | 71.1/ 90.0 |
| AdvProp + AdvBN | 77.3 | 7.4/ 17.2 | 51.4/ 75.3 | 1.8/ 4.2 | 70.5/ 89.7 |
| Cutmix | 78.6 | 1.1/ 3.2 | 39.0/ 62.2 | 0.3/ 1.0 | 64.6/ 85.8 |
| Cutmix + AdvBN | 78.4 | 4.1/ 10.3 | 42.3/ 66.3 | 1.4/ 3.4 | 66.7/ 87.4 |
| AA* | 76.4 | 1.9/ 5.3 | 45.8/ 70.2 | 0.8/ 2.3 | 65.5/ 86.9 |
| AA + AdvBN | 76.5 | 6.3/ 15.6 | 54.6/ 78.3 | 2.9/ 6.4 | 66.4/ 87.3 |
| Fast AA | 77.8 | 1.9/ 5.0 | 43.4/ 67.0 | 0.8/ 2.5 | 66.8/ 87.3 |
| Fast AA + AdvBN | 77.6 | 5.1/ 12.8 | 44.4/ 68.5 | 1.8/ 4.4 | 67.4/ 87.8 |
| AugMix | 77.6 | 3.9/ 9.9 | 53.5/ 77.0 | 1.0/ 2.7 | 71.9/ 90.7 |
| AugMix + AdvBN | 77.8 | 8.6/ 19.5 | 50.9/ 74.7 | 2.4/ 5.3 | 70.2/ 89.7 |

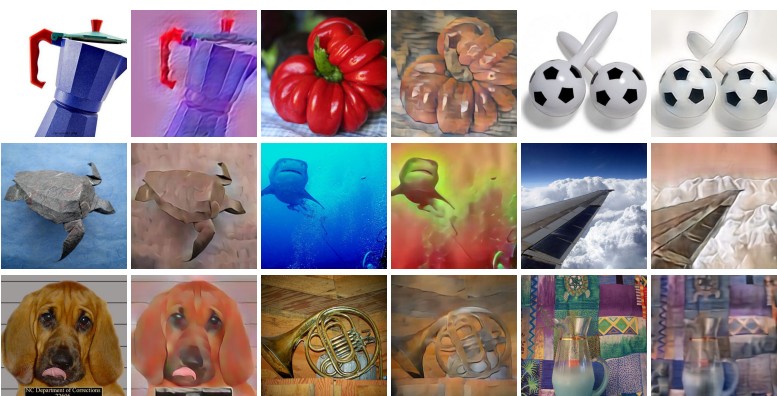

Figure 7: More example images. For each pair of adjacent columns, original versions are on the left, ImageNet-AdvBN-VGG is on the right.

## C   Details concerning the training budget

We evaluate the training time of our method on a workstation with 4 GeForce RTX 2080 Ti GPUs. We use the default settings for AdvBN on ResNet-50: an AdvBN layer placed after the conv2_3 layer, and 20 epochs of fine-tuning with 6-step PGD inside the AdvBN layer. Fine-tuning is conducted on the ImageNet training set, containing 1.3 million images. Training in this setting takes approximately 48 hours, with batch size set to 256. Besides the model size (i.e., the number of model parameters),

Table 9: **Training duration of AdvBN under different model configurations**. $l$ denotes the placement of the AdvBN layer within a ResNet-50, and $m$ is the number of PGD steps.

| Model configuration | $l$=conv2_3 | | | | | | $l$=conv2_3 | $l$=conv3_4 | $l$=conv4_6 |
|---|---|---|---|---|---|---|---|---|---|
| | $m$=3 | $m$=4 | $m$=5 | $m$=6 | $m$=7 | $m$=8 | $m$=6 | | |
| Training duration (hrs) | 30 | 36 | 43 | 48 | 53 | 59 | 48 | 31 | 15 |

there are other factors that can affect the training speed of our method. The first factor is the number of PGD steps used by AdvBN layer, as each step evokes a backpropagation through the later part (after the AdvBN layer) of the network. The default setting of our method on ResNet-50 uses 6 PGD steps, so the training time is longer than standard training for the same number of epochs. Another factor is the placement of AdvBN layer within a network. In each PGD step, gradients only backpropagate through the sub-network after the AdvBN layer, so it takes a notably shorter time to train a model with our method, if the AdvBN layer is placed at later network layers.