# OpenReview forum: "Encoding Robustness to Image Style via Adversarial Feature Perturbations"
_NeurIPS.cc/2021/Conference — NeurIPS 2021 Poster_

### Official Review · Reviewer_WKvM · 2021-07-08

**Rating:** 7
**Confidence:** 4

**Summary:**

This paper propose AdvBN as a variant of batch normalization, to increase the robustness to distributional shifts of existing CNNs.


**Limitations And Societal Impact:**

Yes.

**Main Review:**

The authors propose to improve DNNs' robustness to different image styles by adversarially perturbing the features statistics during training, and fine-tuning the parameters to make them robust to the perturbed features.
The authors implement this idea as a DNN layer named AdvBN, which could be integrated into different DNN architectures.

Strengths:
* The intuition (perturbing feature statistics during fine-tuning to increase the robustness) makes sense.The authors first visualize the effects of perturbing feature statistics by an autocoder, to show that the feature statistics can indeed encode the "style" of different images. Then the authors propose to adversarially modify the feature statistics which could yield generate different image styles, and then make the classifier to be robust to these styles in the training time.
* The authors integrate the proposed AdvBN on five data augmentation methods, and all test cases on four datasets show that the AdvBN can indeed improve the robustness to distribution shifts.
* I particularly like the experiment designed in Section 5.3, which demonstrates that the ResNet-50 model with AdvBN can learn more robust feature representations across different image styles, when compared with a standard ResNet-50 model.

Weakness:
* In Table 1, since the authors perform fine-tuning for AdvBN, the total training time seem to be longer than baseline methods . The number of training steps should be carefully controlled to make a fair comparison.



**Time Spent Reviewing:**

30

---

> ### Author Response · Authors · 2021-08-10
> **Reply to Reviewer WKvM**
>
> We appreciate the reviewer’s positive comments.
> We agree that our models are trained for slightly more steps since we apply AdvBN for fine-tuning existing models.  However, considering that the models are usually trained until convergence, an additional 20 epochs of training does not improve baseline’s performance, especially for AugMix that has been trained for 240 epochs and Cutmix that has already been trained for 300 epochs.  Nonetheless, we have now trained competing models for 20 additional epochs each and will include them in our final version.

---

### Official Review · Reviewer_1irr · 2021-07-15

**Rating:** 7
**Confidence:** 3

**Summary:**

This paper proposes to improve robustness on ImageNet classification by adversarially optimizing the batch-norm statistics of a pretrained classifier. The initial layers of the classifier are frozen, and only the upper subnetwork of the classifier is finetuned after the addition of the adversarial batch-norm layer. The benefits of the method often appear in conjunction with image augmentation methods, where applying AdvBN on top of an image augmentation method tends to result in better performance.

**Limitations And Societal Impact:**

Limitations and impact are addressed.

**Main Review:**

The methodology of this paper is fairly simple, and the results are consistent across the board on various domains (several ImageNet variants, several classification architectures, and on a segmentation task). Ablations show the impact of altering the position of the AdvBN layer and on number of PGD steps. Overall, the paper reads smoothly and is clearly written.

A few points that would be helpful to clarify:
-  in A.3 it is mention that the original BN parameters are used for some datasets, and auxiliary BN is for others. It would be helpful to provide more details on how this choice is made, and full results on using both BN option to see how sensitive the performance is for this choice. Is main or auxiliary BN used for the segmentation experiment?
- in Table 7, why are the reconstructed classifications without adversarial perturbation much worse than clean ImageNet classifications? It would be helpful to see what the reconstructions look like.
- in Eqn 1, the notation $\delta f$ seems a bit unclear. Is it standard deviation of the features f?



**Time Spent Reviewing:**

2.5

---

> ### Author Response · Authors · 2021-08-10
> **Reply to Reviewer 1irr**
>
> We appreciate the reviewer’s positive feedback and valuable suggestions.
> To answer the raised questions:
>
> * In A.3, we mention that our choice is based on the distance between the testing and the training distributions, and we choose to use auxiliary BN for testing data that are far from the training data. We will include the full results of using both BNs and extend the discussion in our updated version of this paper. Thanks for the suggestion. For segmentation, we use the main BN for evaluation, as we mention in line 274, for a consistent setting with the Adv. Training baseline. However, we find the difference between main and auxiliary BN to be minor, with the auxiliary one actually yielding slightly better performance.
>
> * We presume that the degraded performance on reconstructed images is due to our relatively simple VGG-based autoencoder architecture. CNNs have been shown to rely heavily on localized clues, while reconstructed images may lose this information. The reconstructed images can be found in Figure 3, denoted by “m=0”. Note that despite the performance degradation on reconstructed images, the performance on AdvBN generated images is still much lower than that on the reconstructed natural images.
>
> * Yes, this is the standard deviation of the features $f$.  Thanks for bringing this to our attention.  We have changed it to $\delta(f)$ in our updated version.

---

> > ### Comment · Reviewer_1irr · 2021-08-26
> > **Reviewer Response**
> >
> > Thanks to the authors for their response. I will retain my original rating of 7. It would be helpful to highlight the connection between Fig 3(m=0) and table 7 reconstructions in the supplemental, or even better provide more examples like in SM Fig 7 for the AdvBn counterparts.

---

> > > ### Author Response · Authors · 2021-08-27
> > > **Reply to Reviewer 1irr**
> > >
> > > Thanks for pointing this out.  We have now added to our draft the additional examples you suggested, and we are also including plots of the differences between reconstructions and the original images (reconstruction - original).

---

### Official Review · Reviewer_4gjW · 2021-07-16

**Rating:** 6
**Confidence:** 4

**Summary:**

The paper studies the problem of making predictive models robust to distributional shifts. This is achieved using an adversarial training strategy that adversarially perturbs batch norm statistics of a pre-trained model, which effectively captures variations in image style such as color, texture, etc. This worst-case optimization strategy exposes the model to larger variations in image statistics and semantics rather than just just pixel-based perturbations, and as a result is expected to produce better robustness under large distribution shifts.


**Limitations And Societal Impact:**

Yes, the paper addresses some of its weaknesses in the broader problem of robustness under distributional shifts.

**Main Review:**

**Strengths**
+ The paper adds to emerging evidence that BN statistics can play a critical role in designing classifiers that are robust to unknown noisy changes and domain changes.
+ Interesting technique to visualize the adversarially perturbed features, which many adversarial methods are not able to do. This is useful in understanding the kinds of attributes being perturbed during optimization. That the most relevant are color, and texture are also intuitive, corroborating previous studies that argue CNNs are very reliant on these properties for classification.
+ I like the formulation that it is generic (network agnostic), and simple to implement.

**Weaknesses**
+ There are a few concerns with the work with regard to novelty — the idea of using adversarial training to obtain robustness to large semantic shifts instead of pixel-wise perturbations has been explored in the past (Gokhale et al., 2020 https://arxiv.org/abs/2012.01806). More specifically, the idea of adapting BN statistics to obtain robustness to distribution shifts has also been explored (Schneider et al., 2020, https://arxiv.org/abs/2006.16971; Nado et al., https://arxiv.org/abs/2006.10963); and other kinds of layer statistics (Wang et al., https://arxiv.org/abs/2006.10726)  albeit for test time adaptation, which is a different application than those considered in this paper. However, the broad idea that robustness in this manner can be achieved has been somewhat established. This paper is using the same approach in an adversarial training setting.
+ Next, it seems the benchmarks on which robustness is measured are better suited for AdvBN (like stylized imageNet, sketch etc.) which mostly vary along texture and color anyway. On ImageNetC it is clear AdvBN on its own is unable to match the performance of existing baseline techniques like Adv Prop, Augmix etc. Many methods using adv robustness have also considered larger domain shifts (like in a domain generalization setting) which is more challenging, and will perhaps be a more fairer evaluation for AdvBN as well.
+ Most evaluations are performed like ablations, with robustness methods mainly based on augmentation — whereas there are several recent approaches that have shown promise even otherwise which will help provide context to AdvBN’s performance benchmarks. This is the same issue on the semantic segmentation results — no other baselines are reported other than baseline and adv training, which are weak for this task which involves large semantic shifts — advBN is better by design.
+ Question about the approach:  Why is only a single AdvBN layer used? How does the performance vary if more than one AdvBN layers are used or do they end up being redundant?

**Time Spent Reviewing:**

3.5

---

> ### Author Response · Authors · 2021-08-10
> **Reply to Reviewer 4gjW**
>
> We thank the reviewer for their thorough and constructive feedback.
> We would like to address the concerns below.
>
> * Related Work:
> [1] Gokhale et al., Attribute-Guided Adversarial Training for Robustness to Natural Perturbations, 2020.
> [2] Schneider et al., Improving Robustness against Common Corruptions by Covariate Shift Adaptation, 2020.
> [3] Nado et al., Evaluating Prediction-Time Batch Normalization for Robustness under Covariate Shift, 2020.
> [4] Wang et al., TENT: Fully Test-Time Adaptation by Entropy Minimization, 2020.
>
> Thanks for pointing out these works. We have included them in our updated version. In addition, [2] is already included in our submitted paper but with the title from an earlier version (we will update it). However, we do not think this prior work affects the novelty of our method, with explanations provided below.
> Although both [1] and our method adopt the adversarial training strategy, our intuition and methodology differ substantially. [1] proposes to craft perturbations based on predefined attributes and transformations (denoted as “surrogate function” in [1]), both of which are specified by human users and serve as strong priors. For example, to perturb the predefined “shape” attribute of an image, they choose to use AttGAN to generate images with different shapes. Another example is to enhance robustness against image corruptions, they perturb images by adding Gaussian noise and applying Gaussian filters.  Our method, on the other hand, is motivated by the observation that CNNs can naturally encode image “style” information in feature statistics without any specified semantic attributes or hand-crafting. We propose to perturb such statistics directly in the feature space without generating new training samples.
> Concerning [2, 3, 4], we agree that they share the idea of simulating distributional shifts through modifying BN statistics. Moreover, as mentioned in [2,3,4] as well as in our work, the potential effectiveness of Batch Normalization for domain adaptation was proposed in Li et al. (2017).  The methods in [2,3,4] use test samples from a target domain to estimate the target distribution and adapt or optimize the normalization layers in a model using the (known) target statistics. Our method, in contrast, is target-agnostic, works solely at training time, and does not require any knowledge of the target domain. This choice was made because we study domain generalization rather than domain adaptation; i.e., we seek to transfer models to unseen domains that may come along in the future without having to modify our model, rather than to adapt a model to a particular domain after collecting new test data and estimating target distributions on this data.  Because we seek to make models resistant to worst-case unseen domain shifts, our goal is to achieve “domain robustness.”
>
> * Larger domain shifts
>
> Since our method is motivated by literature on image stylization, we agree that our method has an advantage on benchmarks that feature style variations. However, we also show that our method is complementary to existing techniques.
> Concerning domain generalization, we have now tested our method on the PACS dataset. We adopt the commonly used “leave-one-domain-out” protocol for evaluation, which requires training on three domains and testing on the remaining domain, and we average over the four combinations.  Note that our method is not designed for training in multiple domains collaboratively, and we only substitute the ImageNet pre-trained ResNet-50 with ours in this evaluation.  Nonetheless, AdvBN boosts the performance of ResNet-50 from 82.1% accuracy to 85% (mean accuracy) on PACS.  Moreover, AdvBN is compatible with existing domain generalization methods.
>
> * More evaluations
>
> Many robustness techniques (at least on the task of image classification and on benchmarks like ImageNet-C) involve test-time adaptation using samples from the target domain, or training with additional data from multiple domains.   Our comparisons focused on data augmentation methods because these, like AdvBN, do not require samples from the new domains or additional auxiliary datasets. We also consider MoEx, a feature space method, for comparison because it operates in a similar setting to ours.
> Below, we provide additional results in the cross-domain segmentation problem. We implement the MoEx and AdvProp methods for segmentation. We train them for 110 (80 + 30) epochs for both methods to match the training steps with our AdvBN fine-tuned model. We observe that AdvBN typically yields superior results to AdvProp and MoEx in cross-domain settings.
>
>
> |              |            |      |      |  NYC  |        |        |       |      | Highway |        |        |
> |:--------------:|:------------|:------:|------:|:-------:|:--------:|:--------:|:-------:|:------:|:---------:|:--------:|:--------:|
> |              |            | dawn | fog  | night | spring | winter | dawn  | fog  |  night  | spring | winter |
> |              | AdvProp    | 52.3 | 48.9 | 48.0  | **71.9**   | 48.2   | 18.9  | 22.0 |  13.8   | 24.3   | 17.5   |
> |  NYC-spring  | MoEx       | 54.7 | 53.2 | 51.8  | 71.1   | 49.0   | 21.3  | 24.3 |  19.4   | 27.6   | 18.7   |
> |              | AdvBN      | **57.5** | **55.1** | **55.4**  | 66.5   | **52.7**   | **23.8**  | **26.6** |  **25.9**   | **29.8**   | **23.5**   |
> |              |            |      |      |       |        |        |       |      |         |        |        |
> |              | AdvProp    | 30.8 | 24.1 | 20.3  | 23.2   | 21.4   | **64.6**  | 53.5 |  47.2   | 59.0   | 47.6   |
> | Highway-dawn | MoEx       | 32.0 | 27.5 | 27.6  | 29.6   | 26.7   | **64.6**  | **57.2** |  **57.0**   | 61.0   | 51.1   |
> |              | AdvBN      | **34.0** | **31.6** | **29.6**  | **30.7**   | **29.1**   | 64.5  | **57.2** |  **56.4**   | **61.2**   | **53.2**   |
>
> * Using multiple AdvBN layers
>
> We should have included these ablations in our submission, and we are updating our draft accordingly.  The perturbations at successive layers compound and destabilize the training process.

---

> > ### Comment · Reviewer_4gjW · 2021-08-26
> > **Response**
> >
> > Thanks for the detailed response and new experiments, I appreciate the authors time and effort. I agree with the rest of the reviewers -- and as I have pointed out in my review, I like that AdvBN is a simple approach which is model agnostic, and provides robustness out of the box without additional data. There is a general consensus that this method is useful for robustness of some kinds as evidenced by numerous experiments in the paper.
> >
> > My point regarding ImageNet-C is that AdvBN alone is quite a poor baseline compared with other methods (AugMix, Fast AA, AA, etc.) and considering it is model agnostic, it can be combined with existing approaches to boost performance -- which makes it much harder to tease out what the contribution of AdvBN is over combinations of other adversarial robustness methods (for e.g. GUD [42] + CutMix or MADA [https://arxiv.org/pdf/2003.13216.pdf]+ AdvProp etc. etc. and different combinations of these)
> >
> > There also many approaches that *do not use* examples at test time for robustness and generalization. I would point the authors to baseline methods in a recent ICLR 2021 paper (can be considered concurrent, but the baselines are nevertheless useful) https://openreview.net/pdf?id=BVSM0x3EDK6
> >
> > The Domain generalization experiments need more evaluation (its not clear what validation protocol is used by the authors here) since recent domain generalization benchmarks with empirical risk minimization (ERM) -- the naive approach tested here with a basic ResNet-50 is 85.5% averaged on PACS. (see https://github.com/facebookresearch/DomainBed published in ICLR 2020) -- while it is appreciable that there is a boost in performance on the baseline model, we need to be sure that the improvement persists under the right evaluation protocol.
> >
> > This is an interesting paper, but as I have pointed out there some gaps that need to be addressed, and I would like to retain my original review as a result.

---

> > > ### Author Response · Authors · 2021-08-30
> > > **Reply to Reviewer 4gjW**
> > >
> > > Thanks to the reviewer for their response and suggestions. We appreciate the positive comment on our approach and would like to address the raised concerns.
> > >
> > > * Domain generalization experiments
> > > As we mentioned previously, we use the common validation protocol for the PACS dataset which is referred to as the “leave-one-domain-out” protocol in [ [Li et al. (2019)](https://arxiv.org/pdf/1902.00113.pdf), [Zhou et al. (2020)](https://arxiv.org/pdf/2007.03304v3.pdf), [Xu et al. (2021)](https://openaccess.thecvf.com/content/CVPR2021/papers/Xu_A_Fourier-Based_Framework_for_Domain_Generalization_CVPR_2021_paper.pdf)]. In [Gulrajani et al. (2020)](https://openreview.net/pdf?id=lQdXeXDoWtI) (the ICLR version of https://github.com/facebookresearch/DomainBed), we find the naming of this validation protocol to be different from other literature. Nevertheless, we adopt the open-source implementation of Gulrajani et al. (2020) along with our AdvBN pre-trained weights. As a result, we find that AdvBN improves the performance of the ERM method from 85.5% to 86.5% based on their implementation, achieving better performance than any method in Table 3 (top table) of Gulrajani et al. (2020).  In addition, we would like to point out that the ERM model in Gulrajani et al. (2020) uses a number of performance-enhancing tricks such as random grayscale data augmentation and freezing batch normalization during training, and it uses different train/val splits than the defaults in PACS.  These differences explain the gap between our previous result and the one in Gulrajani et al. (2020).
> > > * Other baseline methods in [Xu et al. (2021)](https://openreview.net/pdf?id=BVSM0x3EDK6)
> > > Thanks for pointing this out. Xu et al. (2021) evaluate performance on AlexNet and ResNet-18, so we have now evaluated the [SIN](https://openreview.net/pdf?id=Bygh9j09KX) baseline in our setting on ResNet-50. The results are shown below. Vanilla AdvBN (without advanced data augmentations) consistently outperforms SIN on all datasets.  We have updated our draft to include these results, and we are continuing to run additional baselines.
> > > |    Method    |    ImageNet   |   ImageNet-C  |  ImageNet-Ins.  |  ImageNet-Sketch  |  ImageNet-Style  |
> > > |:------------:|:-------------:|:-------------:|:---------------:|:-----------------:|:----------------:|
> > > |              |Top-1 acc. (↑) |   mCE (↓)     |  Top-1 acc. (↑) |  Top-1 acc. (↑)   |  Top-1 acc. (↑)  |
> > > |    AdvBN     |    **77.0**   |    **72.7**   |    **69.5**     |     **27.9**      |      **11.9**    |
> > > |      SIN     |    76.7       |      73.8     |      68.5       |       26.9        |        10.4      |
> > >
> > > * Combinations of other adversarial robustness methods
> > > We agree that there are numerous possible combinations of methods, and it is intractable to try them all. Instead, we simply show that AdvBN, a new approach, can improve the performance of existing methods over results reported in the literature.

---

> > > > ### Comment · Reviewer_4gjW · 2021-08-30
> > > > **Response**
> > > >
> > > > Hi, thanks for the new evidence on the domain generalization benchmark and comparison with SIN  -- this make a more convincing argument that advBN is a general purpose robustness approach. I am happy to raise my score.

---

### Official Review · Reviewer_316u · 2021-07-17

**Rating:** 6
**Confidence:** 5

**Summary:**

The paper proposes a new method for augmenting training based on worst-case perturbations of batch statistics, batch mean and variance. The idea is that these statistics encode style information thus making a model robust to small changes of their values will promote style invariance. The authors evaluate their method on variants of the ImageNet dataset and compare its performance with existing data augmentation alternatives. They find that while the method itself does not always outperform these alternatives, combining it with them can improve performance.

**Limitations And Societal Impact:**

Yes, they do.

**Main Review:**

The paper studies an important question: how can simple synthetic transformations used during training improve robustness to unforeseen transformations of the data. The proposed method is rather simple and intuitive. At the same time, the performance improvement from using this method alone is too small to justify using it over other methods. Also, the improvement from combining it with other methods is still rather small in most cases. Nevertheless, I still believe that the findings could be of interest to the robustness community.

**Time Spent Reviewing:**

2.5

---

> ### Author Response · Authors · 2021-08-10
> **Reply to Reviewer 316u**
>
> We appreciate the reviewer’s feedback and positive comments.

---

### Decision · Program_Chairs · 2021-09-27

**Decision:**

Accept (Poster)

**Comment:**

After a thorough discussion (particularly with reviewer 4gjW) there are now four reviews that unanimously recommend acceptance.
Thus, I will also recommend acceptance.